# ATTENTION HEAD ENTROPY OF LLMS PREDICTS ANSWER CORRECTNESS

## ABSTRACT

Large language models (LLMs) generate plausible, yet possibly incorrect answers, posing risks in safety-critical settings, such as medical advice. Although both LLM-as-judge and human evaluations are useful, human evaluation is expensive, whereas LLM-as-judge approaches risk introducing additional hidden errors. To address this, we introduce HEAD ENTROPY, a white-box and scalable method that uses the attention patterns inside the model itself to determine the likelihood of a correct answer while generating the answer. Our key insight is that certain attention heads exhibit distinct entropy patterns when the model gives correct versus incorrect answers. Using a sparse logistic regression classifier on per-head entropies, HEAD ENTROPY achieves 0.07–0.15 AUROC improvements over baselines on 5 instruction-tuned LLMs and 3 QA datasets spanning general knowledge, multi-hop reasoning, and medicine. Through Shapley value analysis, we demonstrate that middle-layer attention heads contribute the most to prediction accuracy, providing mechanistic insight into model failure modes. HEAD ENTROPY offers a practical, interpretable, and computationally efficient approach for real-time correctness estimation during LLM deployment.

## 1 INTRODUCTION

Large language models (LLMs) are increasingly being deployed in safety-critical settings, ranging from medical question answering (QA) and clinical decision support to applications in law and scientific discovery (Singhal et al., 2022; 2025; Tu et al., 2024; Yang et al., 2024; Tanno et al., 2025; Zambrano Chaves et al., 2025). However, even state-of-the-art systems can generate fluent, confident yet potentially *incorrect* responses (Wang et al., 2024; Sivarajkumar et al., 2024; Chang et al., 2025). Validating every output with an expert may be costly and impractical at scale, while relying on domain-specific LLMs for validation could introduce unpredictable biases or hidden errors unless those outputs are also reviewed by experts (Chen et al., 2024; Thakur et al., 2024). To address this, we introduce HEAD ENTROPY, a scalable and interpretable white-box method that uses the attention patterns inside the model itself to determine the likelihood of a correct answer while generating the answer. Our key insight is that certain attention heads exhibit distinct entropy patterns when the model generates correct versus incorrect answers. From an information-theoretic perspective, entropy quantifies the amount of expected information content of a random variable (Shannon, 1948). In the transformer self-attention mechanism, each attention head can be interpreted as producing a probability distribution over tokens. Low-entropy heads focus sharply on a few tokens, while high-entropy heads spread attention diffusely, potentially reflecting broader focus or low confidence. At the start of training, attention head entropy is very high and decreases through the course of training (Zhai et al., 2023). We hypothesize that large attention head entropy across heads and layers is a symptom of limited compression of task-relevant information.

We empirically demonstrate this connection between attention head entropy and an LLM's internal information content to introduce HEAD ENTROPY. During a single forward pass, HEAD ENTROPY estimates the LLM's correctness using the attention entropy of individual heads as features. We use the per-head entropies as features to train a sparse, $\ell_1$-regularized logistic regression that produces calibrated correctness probabilities. Using Shapley-value attribution, we show that influential heads are positioned in middle layers. This yields a mechanistic insight by linking specific heads to fail-

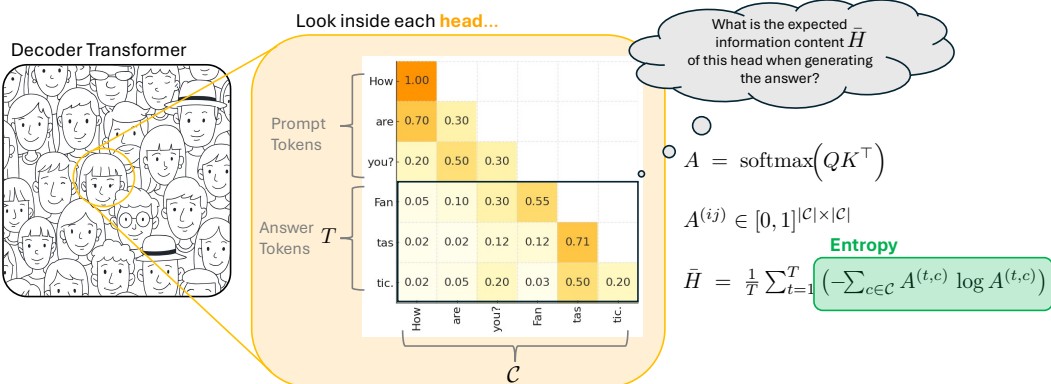

Figure 1: Computing HEAD ENTROPY: We compute HEAD ENTROPY for all $m$ heads and use $\bar{H}_1, ..., \bar{H}_m$ values as features for a logistic regression model to predict the likelihood of a correct answer.

ure modes, providing a simple, practical tool that generalizes across instruction-tuned LLMs with minimal inference overhead.

In summary, we provide

- **quantitative** evidence across 5 instruction-tuned LLM's (thinking and non-thinking) and 3 QA datasets spanning general knowledge, multi-hop reasoning, and medicine, showing that HEAD ENTROPY consistently outperforms established single pass baselines by 0.07-0.15 AUROC and

- **qualitative** evidence that middle layers have predictive information of model correctness, rather than relying solely on distributions over the final token vocabulary. This grounds correctness signals in the model's internal processing, contributing both to more practical deployment strategies and to a deeper scientific understanding of LLMs.

## 2 RELATED WORK

Two fundamental goals motivate this study: (i) *uncertainty estimation* and (ii) *hallucination detection*. Uncertainty estimation provides a measure of how much we can trust a model's output. Hallucination detection, on the contrary, aims to identify statements in generated text that are unsupported or contradict established evidence. Our work translates broader uncertainty estimation and hallucination detection into the more concrete objective of estimating answer-level correctness, that is, the probability that an answer produced by an LLM given an input is correct under a task metric (e.g., exact match, F1).

Prior work differs in how much of the LLM's internal state is used to fulfill these goals: (i) blackbox methods rely only on outputs, often requiring multiple passes or auxiliary models; (ii) graybox methods access shallow signals such as final-layer logits; and (iii) white-box methods leverage deeper representations, such as hidden states or attention maps.

**Black-box methods.** These methods rely solely on the LLM's input and outputs and often require multiple forward passes or auxiliary models, making them expensive and sensitive to domain shift. SelfCheckGPT (Manakul et al., 2023) detects uncertainty via disagreement across sampled completions, while Chain-of-Verification (Dhuliawala et al., 2023) probes consistency through follow-up questions. Other approaches, such as semantic uncertainty (Kuhn et al., 2023), use embedding models to group paraphrases, introducing the reliance on well-aligned external tools. LLM-as-a-Judge methods (Chen et al., 2024; Thakur et al., 2024) delegate evaluation to separate LLMs, potentially introducing biases and weak calibration guarantees. Overall, while these methods are often straightforward to apply to any LLM, they typically trade off efficiency for accuracy and rely on external models that may not generalize well across domains.

**Gray-box methods.** Gray-box methods occupy a middle ground, drawing on limited internal signals such as token log probabilities, top-$k$ logits, or final hidden states. By doing so, they enable single-pass confidence estimation, which makes them attractive in resource-constrained settings. For example, predictors trained over logits can temper the well-documented tendency of LLMs toward overconfidence (Groot & Valdenegro-Toro, 2024), while filters based on final-layer statistics can anticipate distributional shifts that undermine reliability (Pouget et al., 2025). Likewise, conformal prediction methods extend log-probability signals into calibrated sets of plausible completions (Angelopoulos et al., 2021), offering a principled notion of coverage. Despite these strengths, gray-box approaches remain constrained by their reliance on final-layer information, which often calibrates poorly across domains and reduces interpretability to heuristics like "low log-probability" (Geng et al., 2024).

**White-box methods.** These methods leverage richer internal signals such as layer-wise hidden states, attention maps, or neuron activations, often improving accuracy while introducing additional design choices. Several works (Azaria & Mitchell, 2023; Kadavath et al., 2022) have demonstrated that an LLM's hidden layers encapsulate latent knowledge about true and false outputs. (Snyder et al., 2024) employed gated recurrent units (GRUs) and multilayer perceptrons (MLPs) over internal representations to predict correctness. LLM-Check (Sriramanan et al., 2024) combines multiple internal signals, including attention statistics, to construct a correctness metric. These methods often achieve moderate ROC-AUC scores, but are challenging to interpret and sometimes require processing large, high-dimensional tensors, making them costly to deploy in real-time. Even though our approach also relies on internal signals, it focuses on single-pass efficiency, interpretability, and simplicity. Our goal is to achieve better separability between correct and incorrect answer, better interpretability and efficient deployment.

**Interpretability.** Interpretability plays a central role in both safety-critical applications and technical understanding of LLMs. In high-stakes domains, identifying why a model fails is essential for building trust and accountability (Hardt et al., 2016; Obermeyer et al., 2019; Bender et al., 2021). Beyond safety, interpretability also supports technical objectives: understanding which internal components drive behavior can inform model editing, pruning, and targeted fine-tuning. Shapley-value methods, such as SHAP (Lundberg & Lee, 2017), have been widely used to attribute model decisions to specific features. In the context of LLMs, emerging work explores attributing predictions to internal mechanisms, such as attention heads or hidden states, to enable more grounded analysis and interventions.

# 3 BACKGROUND

## 3.1 TRANSFORMER ARCHITECTURE

The classical decoder-only transformer architecture consists of stacked layers, each containing a multi-head self-attention mechanism with a causal mask followed by a token-wise feed-forward (FF) network. Self-attention computes weighted interactions between all tokens in the sequence, allowing the model to capture long-range dependencies. Each attention head learns a distinct pattern of interactions, and the outputs from multiple heads are concatenated. The subsequent FF network applies nonlinear transformations to these attention outputs, enabling higher-order feature interactions. The final layer projects the contextualized token representations into the vocabulary space, producing the token logits used for prediction.

## 3.2 SHANNON ENTROPY

Shannon entropy provides a measure of expected information content in a random variable. For a discrete random variable $X$ with probability mass function $p(x)$, the Shannon entropy is defined as $H(X) = -\sum_x p(x) \log p(x)$. Higher entropy corresponds to greater unpredictability, while lower entropy indicates more certainty. In the context of the transformer architecture, each head can be interpreted as providing a probability distribution as the output of the softmax. The entropy describes the amount of expected information in $X$.

## 4 METHOD: HEAD ENTROPY

HEAD ENTROPY encompasses three components: 1) computing the per-head attention entropies, 2) using a logistic regression model to find predictive patterns, and 3) using per-head Shapley values for interpretation.

### 4.1 COMPUTING HEAD ENTROPY

Motivated by question-answering using instruction tuned LLMs, we consider an autoregressive transformer, $LLM$, with causal masking. The $LLM$ has $m$ attention heads in total, across all layers. We index heads by $k \in \{1, \ldots, m\}$. Let $T$ be the sequence length.

Given an input sequence $x$ (a tokenized question), the model outputs a prediction $\hat{y}$ (a tokenized answer) at nearly deterministic decoding (temperature $\approx 0$):

$$\hat{y} = LLM(x). \tag{1}$$

For a labeled question answering dataset, $\{(x_i, y_i)\}_{i=1}^N$, we define

$$z_i := \mathbb{I}[\hat{y}_i = y_i] \in \{0, 1\}, \tag{2}$$

where equality is measured with an evaluation metric (exact match and F1 score in our experiments). Each attention head $k$ produces

$$A_k = \text{softmax}(QK^\top) \in \mathbb{R}^{T \times T}, \tag{3}$$

where each row of $A_k$ is a probability distribution (i.e., rows sum to 1). The full tensor shape for all $m$ heads is $A \in \mathbb{R}^{m \times T \times T}$. For each head $k$ and token $t$, we compute the Shannon entropy:

$$H_k^{(t)} = -\sum_{j=1}^{T} A_k^{(t,j)} \log A_k^{(t,j)}, \tag{4}$$

to measure the expected information content of the attention mechanism (Shannon, 1948). This quantifies how concentrated or diffuse the attention is, with higher entropy indicating more uncertainty in which past tokens are being attended to. Since entropy is the fundamental measure of uncertainty in information theory, it serves as a principled proxy for interpretability and confidence in attention during generation. We then average over the tokens of a semantic section (e.g., question, thinking, or answer) of length $T_{\text{section}}$:

$$\bar{H}_k = \frac{1}{T_{\text{section}}} \sum_{t=1}^{T_{\text{section}}} H_k^{(t)}, \tag{5}$$

where each entry, $\bar{H}_k$, corresponds to the HEAD ENTROPY of one attention head, where $k \in \{1, \ldots, m\}$. This yields a fixed-dimensional feature vector for each example $i$,

$$\bar{H}_i = [\bar{H}_{1,i}, \ldots, \bar{H}_{m,i}]^\top \in \mathbb{R}^m. \tag{6}$$

In addition to improving interpretability, section-level aggregation reduces memory and compute overhead compared to token-level entropy analysis, collapsing the cost to constant terms per example for a fixed number of attention heads $m$. This makes large-scale or real-time analysis more practical while preserving alignment with the task's semantic structure.

### 4.2 FEATURE SELECTION AND CLASSIFICATION

We use fixed-dimensional feature vector for each example $i$, $\bar{H}_i \in \mathbb{R}^m$ to train an $\ell_1$-regularized logistic regression classifier

$$f : \mathbb{R}^m \to [0, 1] \tag{7}$$

to estimate the probability of correctness:

$$\mathbb{P}(z_i = 1 \mid \bar{H}_i) = f(\bar{H}_i) = \sigma(\beta^\top \bar{H}_i + b), \tag{8}$$

where $\beta \in \mathbb{R}^m$ are the coefficients, $b$ is the bias term and $\sigma$ is the logistic sigmoid. Any classifier can be used here; we chose logistic regression due to simplicity and interpretability.

### 4.3 COMPUTATIONAL EFFICIENCY AND COMPLEXITY

As HEAD ENTROPY essentially uses summary statistics on the existing attention patterns the computational footprint is minimal compared to that of LLM inference.

**Feature generation (entropy).** For each (head, query), let $x$ be the attention logits over keys and $p = \text{softmax}(x)$ with $Z = \sum_j e^{x_j}$. Using the identity

$$H(p) = \log Z - \langle p, x \rangle$$

(with log-sum-exp stabilization), computing entropy requires one log-sum-exp and one dot product per query, reusing quantities already computed for softmax. Thus the asymptotic cost matches the softmax step: with $m$ heads, sequence length $T$, and an attention window of size $T_{\text{section}}$, the work is $O(mTT_{\text{section}})$ with only a small constant-factor overhead relative to the usual softmax. In full attention, this is $O(mT^2)$.

**Memory.** If features are computed on the fly and only per-head aggregates are retained, additional memory is $O(m)$ rather than $O(mTT_{\text{section}})$ for storing full attention maps, since maps are not needed elsewhere.

**Classification.** The logistic regressor consumes one feature per head (dimension $m$), so inference is $O(m)$ per example and negligible compared to attention. It does not scale with $T$ once the features are aggregated.

**Training runtime.** Training the logistic layer is fast; end-to-end time is dominated by generating features from model forward passes.

Overall, the extra cost of HEAD ENTROPY is strictly less than running a second forward pass, and the additional memory needed is only proportional to the number of attention heads ($O(m)$), rather than the full attention maps.

### 4.4 EXPLAINING PREDICTIONS WITH SHAPLEY VALUES

To attribute predictive performance to specific heads, we compute Shapley values (Shapley, 1953; Lundberg & Lee, 2017) on the trained classifier:

$$\phi_k(f) = \frac{1}{m!} \sum_\pi \left[ f(P_k^\pi \cup \{k\}) - f(P_k^\pi) \right], \tag{9}$$

where $\pi$ ranges over all feature orderings, $P_k^\pi$ is the set of features preceding $k$ in $\pi$, and $\phi_k(f)$ quantifies the contribution of head $k \in \{1, ..., m\}$. Using Shapley values, we can construct a heat map over heads to visualize which heads contribute positively (red) or negatively (blue) to the prediction of being correct (Appendix Fig. 12d).

High-level pseudocode can be found in Method 1.

---

**Method 1** Head-Entropy: Based Outcome Prediction

---

**Require:** Dataset $\{(x_i, y_i)\}_{i=1}^N$; Transformer with $m$ heads; sequence length $T$ of $x_i$
1: **for** each $(x_i, y_i)$ **do**
2:      Obtain $\hat{y}_i$ and attention tensor $A_i \in \mathbb{R}^{m \times T \times T}$ with Eq. 3
3:      **for** each head $k$ and token $t$ **do**
4:          Compute entropy $H_{k,i}^{(t)}$ with Eq. 4
5:      **end for**
6:      Average entropies over section tokens $\rightarrow \bar{H}_{k,i} \in \mathbb{R}$ like Eq. 5
7:      Stack to form feature vector $\bar{H}_i = [\bar{H}_{1,i}, ..., \bar{H}_{m,i}] \in \mathbb{R}^m$
8: **end for**
9: Stack $\bar{H}_i$ to form feature matrix $\bar{H} \in \mathbb{R}^{N \times m}$
10: Define labels $z_i = \mathbb{I}[\hat{y}_i = y_i]$
11: Train $\ell_1$-regularized logistic regression $f$ on $\bar{H}_i$ to predict $z_i$
12: **return** Classifier $f$

---

| | TriviaQA | HotpotQA | MedMCQA |
|---|---|---|---|
| # Training Examples | Sampled 50k / 130k | Sampled 50k / 98k | Sampled 50k / 183k |
| # Validation Examples | 17,944 | 7,405 | 4,183 |
| Task | Open-domain QA | Multi-hop QA | Multiple-choice QA |
| Domain | General | General | Biomedical |

Table 1: Datasets used in experiments, with statistics on training/validation splits and model-specific answer lengths.

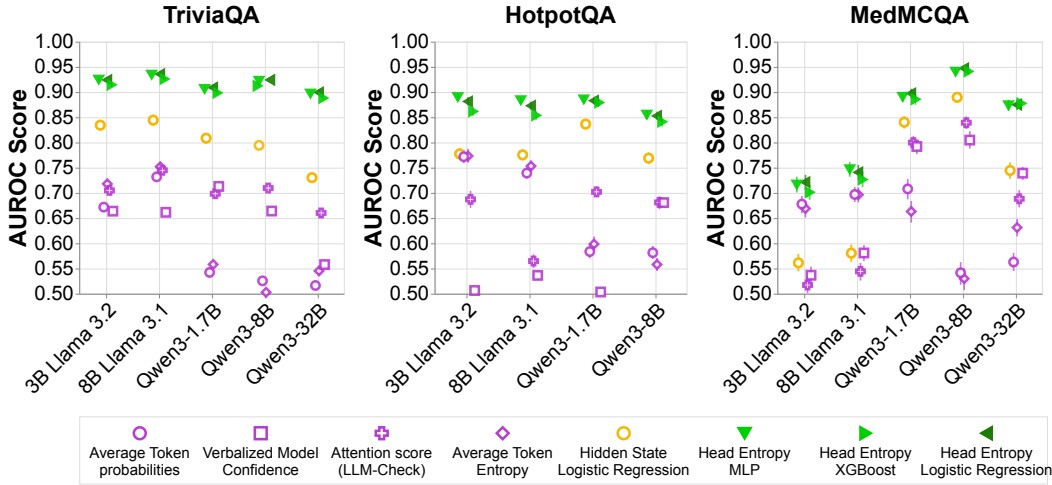

Figure 2: Methods highlighted in pink utilize the model's raw statistics directly. Yellow methods use embeddings from the LLM followed by a trained logistic regression classifier. The green method leverages HEAD ENTROPY features and also trains a logistic regression/MLP/XGBoost model. Confidence intervals (bootstrapped) are smaller than the symbols. See Appendix for exact values.

## 5 EXPERIMENTAL SETUP

**Baselines.** We compare our method against several single–forward pass approaches. The first baseline is token probability, defined as the average token probability over the generated section. We also include verbalized certainty, where the model explicitly states its own confidence inline, following Kadavath et al. (2022) (prompt details are provided in Appendix A.1). Another comparison is the attention score method (LLM-Check, layer 20), which uses a kernel similarity map of self-attention across different tokens introduced by Sriramanan et al. (2024). We further evaluate token entropy, the average entropy of the output token distribution, and hidden state regression, a linear regression on final-layer hidden states similar to the method proposed by Xie et al. (2024).

**Models.** Our experiments cover both reasoning-oriented and general-purpose instruction-tuned LLMs. For reasoning-oriented models, we evaluate the Qwen3 family at 1.7B, 8B, and 32B parameters, enabling us to examine how HEAD ENTROPY varies across model scales. For general-purpose instruction-tuned models, we include Llama 3.1 (8B) and Llama 3.2 (3B). All evaluations are performed with default decoding parameters, corresponding to nearly greedy decoding (temperature $< 0.001$).

**Datasets.** We conduct experiments on three standard yet diverse QA benchmarks: TriviaQA (Joshi et al., 2017), a general question-answering dataset; HotpotQA (Yang et al., 2018) (distractor setting), a multi-hop question-answering dataset with long context; and MedMCQA (Pal et al., 2022) a domain-specific multiple-choice dataset. From each dataset, we sampled 50k examples for training and used the validation split for evaluation (Table 1). Examples and prompts for each dataset are provided in Appendix A.3, A.2, A.1. After generation of the answer, we construct a binary label for each example (Eq. 2) with the standard metrics of TriviaQA and HotpotQA; exact match and F1 score ($> 0.5$).

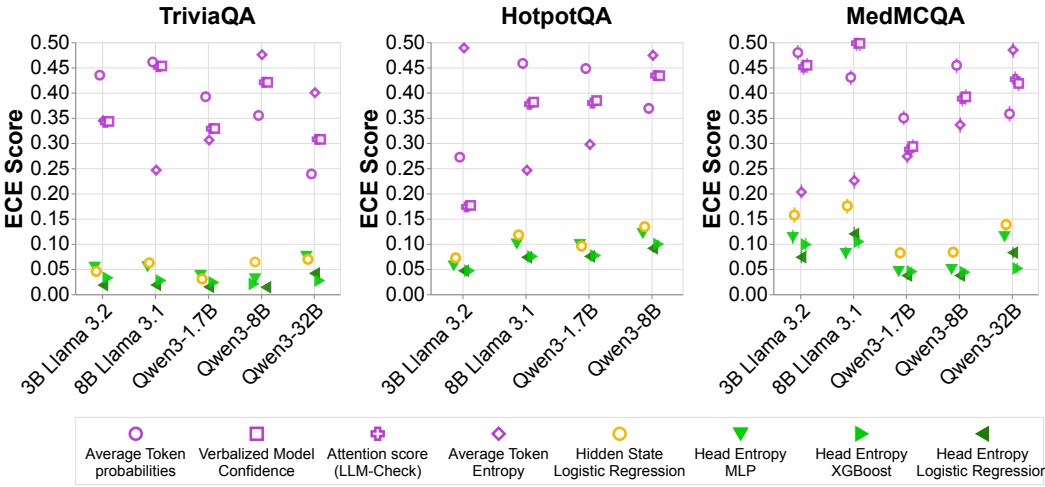

Figure 3: Expected calibration error (ECE) of different methods across datasets (lower better). Logistic regression and related models achieve strong calibration overall ($<0.1$) (yellow and green), with HEAD ENTROPY methods providing the lowest error. Confidence intervals (bootstrapped) are smaller than the symbols. See Appendix for exact values.

s

**Evaluation Metrics.** We report three sets of metrics. First, we assess interpretability through qualitative attribution, using Shapley values computed with SHAP to identify the influential heads and their location within the model. Knowing whether early, middle, or later layers dominate decision-making highlights which representations are portable across models and which are task-specific, guiding interoperability and modular reuse. Second, we measure how well HEAD ENTROPY and the baselines separate the positive and negative classes using AUROC for each semantic slice (question, thinking, answer). Third, we evaluate calibration via Expected Calibration Error (ECE, 30 bins). Confidence Intervals (CIs) are estimated via bootstrapping ($B$=1000).

## 6 EXPERIMENTS AND RESULTS

We evaluate HEAD ENTROPY by comparing against established single-pass baselines across five instruction-tuned LLMs (Qwen and Llama families) and three QA datasets (TriviaQA, HotpotQA, MedMCQA). Our results are organized to first describe the overall performance before analyzing robustness across models, datasets, and ablations.

### 6.1 OVERALL COMPARISON TO BASELINES

Across all models and datasets, HEAD ENTROPY statistically significantly outperforms the strongest single-pass baselines by **0.07–0.15 AUROC** while achieving the lowest calibration error (**ECE $<$ 0.10**). Figure 2 shows aggregate AUROC scores, and Figure 3 shows calibration errors, with 95% bootstrap confidence intervals. These results establish HEAD ENTROPY as both *more discriminative* and *better calibrated* than token-probability, verbalized certainty, attention-score, token-entropy, and hidden-state regression baselines.

### 6.2 BREAKDOWN BY MODEL FAMILY AND SIZE

To test robustness across architectures and scales, we evaluate HEAD ENTROPY on the Qwen3 family (1.7B, 8B, 32B) and the Llama family (3B, 8B)( Figure 2).

**Stable across scales.** For Qwen3, performance with HEAD ENTROPY varies by at most 0.07 AUROC across model sizes, with overlapping confidence intervals. For Llama, the variation is even smaller ($\leq 0.03$ AUROC). In contrast, baselines such as hidden-state logistic regression and

attention-score show swings of 0.10–0.15 AUROC across scales, highlighting their sensitivity to parameter count.

**Consistent across families.** Across the Llama and Qwen3 families, HEAD ENTROPY exhibits stable performance on TriviaQA and HotpotQA, with differences limited to within 0.03 AUROC with overlapping CIs. On MedMCQA, however, the gap between model families widens substantially, reaching 0.20 AUROC. Notably, all baselines also show a relative drop in performance for Llama models on MedMCQA. For other approaches, such as Average Token Probability, HEAD ENTROPY delivers higher scores on Llama models compared to Qwen models, highlighting family-specific differences.

## 6.3 BREAKDOWN BY DATASET AND DOMAIN

We next compare performance across the three QA datasets: TriviaQA (general knowledge), HotpotQA (multi-hop reasoning), and MedMCQA (domain-specific medical multiple-choice) ( Figure 2). Across datasets with very different characteristics, single-hop questions (TriviaQA), long multi-hop reasoning (HotpotQA), and medical domain knowledge (MedMCQA), HEAD ENTROPY consistently outperforms single-pass baselines, though absolute performance varies with task difficulty.

**General QA datasets.** On TriviaQA and HotpotQA, HEAD ENTROPY consistently outperforms all baselines by 0.07–0.15 AUROC despite the substantial in average question length (26 and 1243 tokens, respectively)(Appendix Table 3). The relative ranking of baselines is stable across the two datasets, with token-probability and hidden-state regression typically the strongest among them.

**Domain-specific QA.** On MedMCQA, overall AUROC scores are lower for all methods, reflecting the increased difficulty of the medical domain. Nonetheless, HEAD ENTROPY retains a positive margin over baselines, with statistically significant gains of 0.05 AUROC.

## 6.4 ABLATIONS AND VARIANTS

We now analyze how different design choices affect HEAD ENTROPY. Answer-token entropies are the most reliable predictors of correctness, but think-token entropies provide early signals before the final answer is produced. Cross-dataset experiments indicate that while some patterns transfer, combining datasets during training is more robust than relying on transfer from a single domain.

**Question vs. Think vs. Answer tokens.** We apply HEAD ENTROPY separately to attention entropies computed over the input question, intermediate "think" tokens, and final answer tokens (Figure 4). Answer-token features are the most predictive overall, with AUROC typically 0.10–0.30 higher than question- or think-token features. Between the earlier stages, think tokens provide modestly stronger signal than question tokens, suggesting that intermediate reasoning adds diagnostic value, though with lower separability than final answers.

**Cross-dataset transfer.** We train the logistic regression classifier on one dataset and evaluate on another (Table 2). Performance drops by 0.05–0.35 AUROC compared to in-domain training, indicating that dataset-specific patterns matter. However, training jointly on all datasets yields performance close to in-domain models, suggesting that HEAD ENTROPY features capture signals with at least partial transferability.

## 6.5 CALIBRATION AND RELIABILITY

Beyond separability, it is important that predicted probabilities are well calibrated. Figure **??** reports expected calibration error (ECE, 30 bins) across datasets.

**Calibration performance.** All methods that train a logistic regressor over internal features (hidden states or entropies) achieve substantially better calibration than raw-output baselines, with ECE typically below 0.17. HEAD ENTROPY achieves the statistically significant lowest calibration error overall, often well below 0.10. This indicates that correctness probabilities from HEAD ENTROPY can be interpreted directly and used to rank predictions by reliability. Well-calibrated outputs allow practitioners to identify examples where the model is likely to be incorrect, without needing ground-truth labels. (Appendix 9)

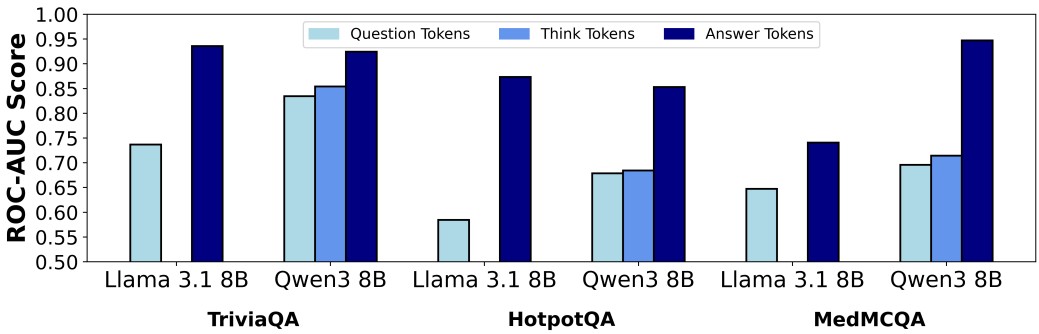

Figure 4: HEAD ENTROPY applied to question and think tokens shows predictive value, but remains consistently weaker than using answer tokens.

### 6.6 QUALITATIVE ANALYSIS: INTERPRETABILITY

Finally, we examine which attention heads contribute most strongly to HEAD ENTROPY predictions. We compute Shapley values of the logistic regression classifier trained on per-head entropies (Appendix Fig. 12d). Beyond quantitative gains, HEAD ENTROPY reveals that mid-layer heads carry the strongest correctness signal, offering interpretable cues that could support monitoring, pruning, or adaptation in deployment.

**Layer-wise patterns.** Averaging absolute Shapley values across layers shows that middle-layer heads are most predictive, whereas early and late layers tend to contribute less. This pattern is consistent across model families (Qwen, Llama) and datasets, suggesting a general role for mid-layer representations in encoding correctness-related signals.

Table 2: Generalization to other datasets (ROC-AUC and relative drop from in-domain training)

| Evaluated on | Trained on | Qwen3 8B (AUROC) | Change (%) | LLaMA 3.1 8B (AUROC) | Change (%) |
|---|---|---|---|---|---|
| TriviaQA | TriviaQA | 0.924 | 0.0 | 0.936 | 0.0 |
|  | HotpotQA | 0.780 | -15.6 | 0.779 | -16.8 |
|  | MedMCQA | 0.881 | -4.7 | 0.607 | -35.1 |
|  | All datasets | 0.917 | -0.8 | 0.924 | -1.3 |
| HotpotQA | HotpotQA | 0.853 | 0.0 | 0.873 | 0.0 |
|  | TriviaQA | 0.709 | -16.9 | 0.709 | -18.8 |
|  | MedMCQA | 0.698 | -18.2 | 0.569 | -34.8 |
|  | All datasets | 0.839 | -1.6 | 0.863 | -1.1 |
| MedMCQA | MedMCQA | 0.947 | 0.0 | 0.733 | 0.0 |
|  | TriviaQA | 0.854 | -9.8 | 0.593 | -19.1 |
|  | HotpotQA | 0.841 | -11.2 | 0.617 | -15.8 |
|  | All datasets | 0.874 | -7.7 | 0.724 | -1.2 |

## 7 CONCLUSION

We introduce HEAD ENTROPY, a white-box method for predicting LLM correctness by analyzing attention head entropy during inference. Our approach achieves ROC-AUC scores of 0.73-0.93 across diverse models and datasets, consistently outperforming baselines by 0.07-0.15 AUROC while maintaining strong calibration (ECE < 0.1). Through Shapley value analysis, we demonstrate that predictive power concentrates in middle-layer attention heads, providing mechanistic insight into model failures. HEAD ENTROPY operates with a single forward pass, improves interpretability, and is based on lightweight logistic regression, making it promising for real-time deployment in safety-critical applications that would otherwise require domain-specific LLMs (Ostmeier et al., 2024; Li et al., 2024), or extensive human expert evaluation (Van Veen et al., 2024; Aali et al., 2025).

## 8 REPRODUCIBILITY STATEMENT

All datasets used in this paper are publicly available. We have added links to the exact versions employed in our experiments to the dataset references. The complete set of prompts utilized in our evaluations is included in the Appendix. To facilitate reproducibility of the reported quantitative results, we will release our source code, along with instructions for dataset preprocessing, model training, and evaluation.

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

# APPENDIX

# CONTENTS

## A  PROMPTS

### A.1  TRIVIAQA PROMPT

```python
def build_triviaqa_prompt(item):
    question = item["question"]

    system_message = (
        "You are a trivia expert. Please answer questions in exactly
            this format:\n"
        "Answer: [1-3 words only]\n"
        "Certainty: [0-100]\n\n"
    )

    user_prompt = f"Question: {question}"

    return system_message, user_prompt
```

### A.2  MEDMCQA PROMPT

```python
def build_ind_prompt(item):

    system_message = (
        "You are a medical expert. Please answer questions in exactly
            this format:\n"
        "Answer: [repeat correct option]\n"
        "Certainty: [0-100]\n\n"
    )
    user_prompt = (
        "Question:"
        + item["question"].split(". ")[-1]
        + "\n"
        + "Options:"
        + " ".join(str(item[op]) + "\n"
                    for i, op in enumerate(["opa", "opb", "opc", "opd"
                        ]))
    )
    return system_message, user_prompt
```

### A.3  HOTPOTQA PROMPT

```python
def build_hotpot_prompt(item):
    system_message = (
        "You are a helpful assistant. "
        "Answer the question using the information in the provided
            passages."
        "Please answer questions in exactly this format:\n"
        "Answer: [1-5 words only]\n"
```

```
 7            "Certainty: [0-100]\n\n"
 8        )
 9        list_sentences = []
10        for sentence_list in item["context"]["sentences"]:
11            list_sentences.extend(sentence_list)
12
13        context_text = "\n".join(sentence for sentence in list_sentences)
14
15        user_prompt = (
16            "Context: "
17            f"{context_text}\n\n"
18            f"Question: {item['question']}\n"
19        )
20
21        return system_message, user_prompt
```

# B   CONTRIBUTIONS OF ENTROPY ACROSS LAYERS

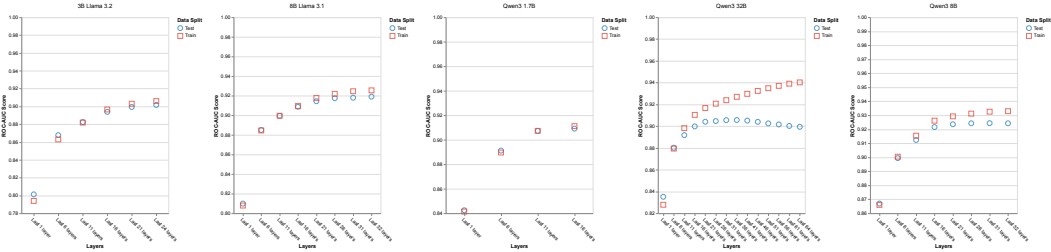

Figure 5: Trivia QA: Layer ablation, where layer 1 is the last layer and then adding more layers

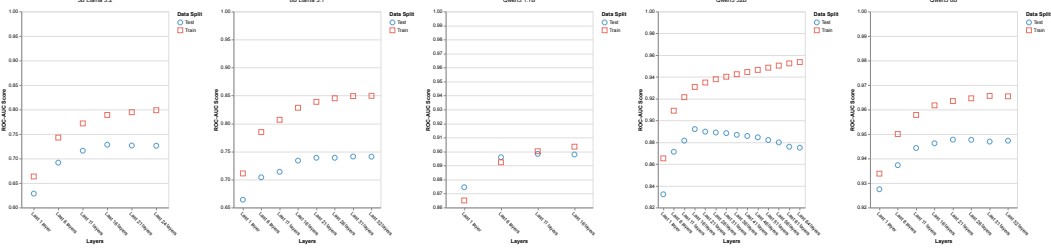

Figure 6: MedMC QA: Layer ablation, where layer 1 is the last layer and then adding more layers

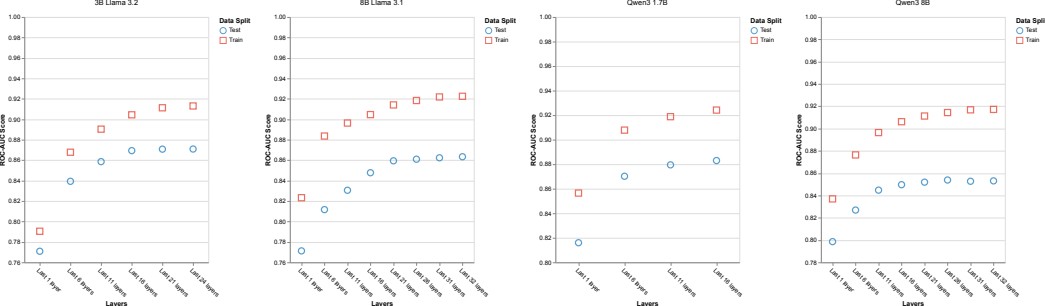

Figure 7: Hotpot QA: Layer ablation, where layer 1 is the last layer and then adding more layers

## C   TOP FEATURES BASED ON HIGHEST SHAPLEY VALUE

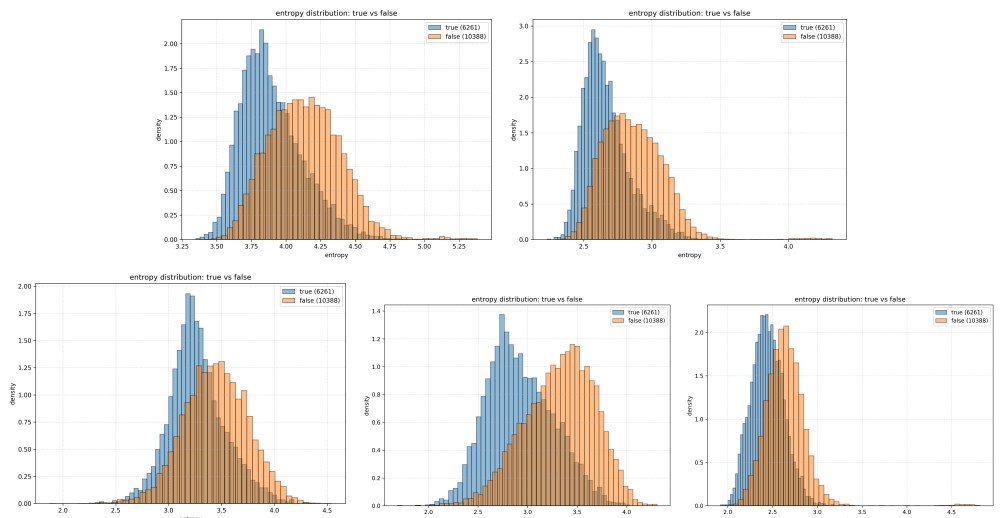

Figure 8: Qwen models 1.7 Billion parameters (excluding overthought examples): All Entropy values are positive and the false examples show a higher variance and higher mean entropy across all validation examples in the TriviaQA dataset

## D   RANKING

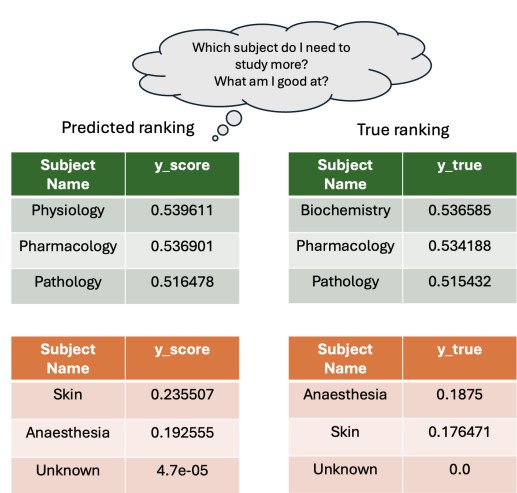

Figure 9: Ranking based on Logistic regression model output matches the ranking when using the ground truth labels for the Qwen3-8 8B model on the Indian Medical exam dataset

## D.1 ASSOCIATION BETWEEN THE MODEL'S OVERALL TEST ACCURACY ON A GIVEN DATASET AND THE ROC-AUC SCORES

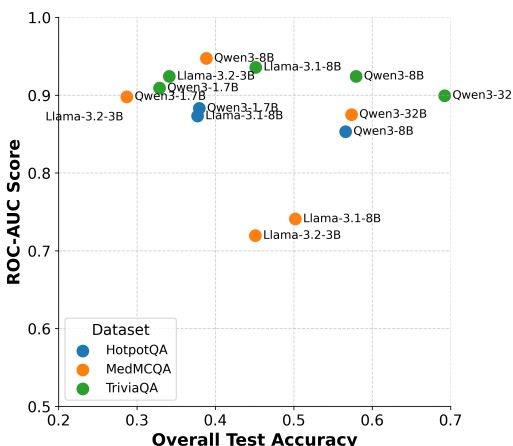

Figure 10: There is no visible association between model performance and predictor performance

## E   F1 SCORE $> 0.5$ RESULTS

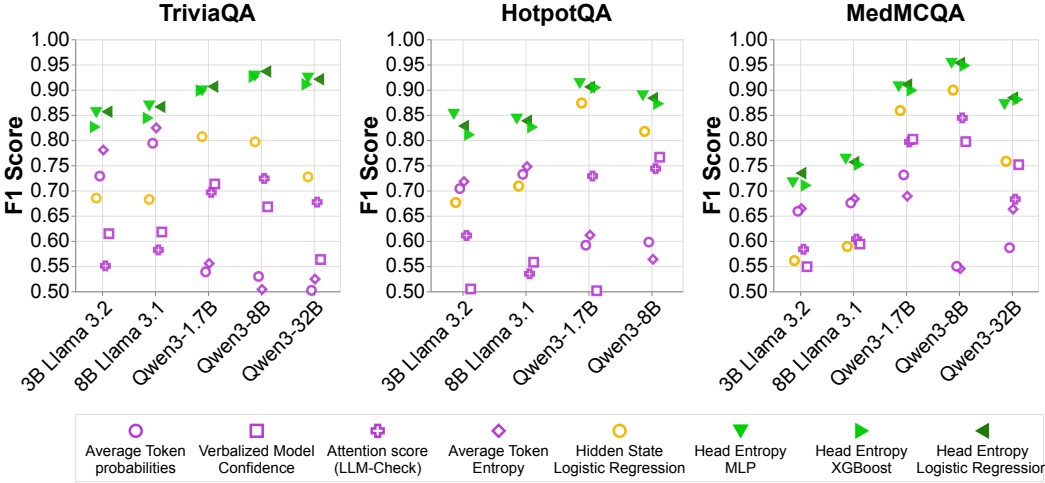

Figure 11: Comparison of single forward pass measures across datasets (higher better), highlighting the superior performance of HEAD ENTROPY models using the F1 score $> 0.5$ as a binary label.

## F  TOKEN LENGTHS AND OVERALL ACCURACY

| Model | TriviaQA | | | | HotpotQA | | | | MedMCQA | | | |
|---|---|---|---|---|---|---|---|---|---|---|---|---|
| | **Acc.** | **Q** | **Think** | **Ans** | **Acc.** | **Q** | **Think** | **Ans** | **Acc.** | **Q** | **Think** | **Ans** |
| Qwen3-1.7B | 0.33 | 26 | 310 | 7 | 0.38 | 1243 | 316 | 19 | 0.29 | 47 | 519 | 6 |
| Qwen3-8B | 0.58 | 26 | 283 | 7 | 0.57 | 1242 | 291 | 8 | 0.39 | 47 | 505 | 6 |
| Qwen3-32B | 0.69 | 26 | 197 | 7 | – | – | – | – | 0.57 | 47 | 368 | 9 |
| Llama-3.2-3B | 0.34 | 25 | – | 8 | 0.17 | 1243 | – | 48 | 0.43 | 46 | – | 12 |
| Llama-3.1-8B | 0.45 | 25 | – | 9 | 0.38 | 1245 | – | 48 | 0.50 | 46 | – | 11 |

Table 3: Performance and token lengths per dataset. Accuracy (using exact match) (**Acc.**), Question (**Q**), Think chunk (**Think**), and Answer (**Ans**) token length.

## G  CONFIDENCE INTERVALS FOR FIGURE 2

Table 4: AUROC Results by Dataset, Model, and Method

| AUROC | AUROC_CI_Low | AUROC_CI_High | Dataset | Model | Method |
|---|---|---|---|---|---|
| 0.7320 | 0.7249 | 0.7397 | TriviaQA | 8B Llama 3.1 | Average Token probabilities |
| 0.6616 | 0.6562 | 0.6670 | TriviaQA | 8B Llama 3.1 | Model confidence |
| 0.7450 | 0.7376 | 0.7516 | TriviaQA | 8B Llama 3.1 | Attention score (LLM-Check) |
| 0.7520 | 0.7453 | 0.7593 | TriviaQA | 8B Llama 3.1 | Average Token Entropy |
| 0.8444 | 0.8385 | 0.8497 | TriviaQA | 8B Llama 3.1 | Hidden State Logistic Regression |
| 0.9348 | 0.9313 | 0.9382 | TriviaQA | 8B Llama 3.1 | Head Entropy MLP |
| 0.9263 | 0.9228 | 0.9300 | TriviaQA | 8B Llama 3.1 | Head Entropy XGBoost |
| 0.9359 | 0.9326 | 0.9392 | TriviaQA | 8B Llama 3.1 | Head Entropy Logistic Regression |
| 0.6719 | 0.6638 | 0.6795 | TriviaQA | 3B Llama 3.2 | Average Token probabilities |
| 0.6639 | 0.6571 | 0.6704 | TriviaQA | 3B Llama 3.2 | Model confidence |
| 0.7051 | 0.6972 | 0.7131 | TriviaQA | 3B Llama 3.2 | Attention score (LLM-Check) |
| 0.7181 | 0.7110 | 0.7254 | TriviaQA | 3B Llama 3.2 | Average Token Entropy |
| 0.8345 | 0.8288 | 0.8402 | TriviaQA | 3B Llama 3.2 | Hidden State Logistic Regression |
| 0.9251 | 0.9214 | 0.9287 | TriviaQA | 3B Llama 3.2 | Head Entropy MLP |
| 0.9147 | 0.9110 | 0.9187 | TriviaQA | 3B Llama 3.2 | Head Entropy XGBoost |
| 0.9242 | 0.9208 | 0.9277 | TriviaQA | 3B Llama 3.2 | Head Entropy Logistic Regression |
| 0.5424 | 0.5332 | 0.5509 | TriviaQA | Qwen3-1.7B | Average Token probabilities |
| 0.7130 | 0.7063 | 0.7198 | TriviaQA | Qwen3-1.7B | Model confidence |
| 0.6984 | 0.6907 | 0.7060 | TriviaQA | Qwen3-1.7B | Attention score (LLM-Check) |
| 0.5582 | 0.5487 | 0.5669 | TriviaQA | Qwen3-1.7B | Average Token Entropy |
| 0.8087 | 0.8027 | 0.8148 | TriviaQA | Qwen3-1.7B | Hidden State Logistic Regression |
| 0.8087 | 0.8027 | 0.8148 | TriviaQA | Qwen3-1.7B | Hidden State Logistic Regression |
| 0.9065 | 0.9023 | 0.9109 | TriviaQA | Qwen3-1.7B | Head Entropy MLP |
| 0.8987 | 0.8943 | 0.9031 | TriviaQA | Qwen3-1.7B | Head Entropy XGBoost |
| 0.9092 | 0.9052 | 0.9136 | TriviaQA | Qwen3-1.7B | Head Entropy Logistic Regression |
| 0.5164 | 0.5072 | 0.5252 | TriviaQA | Qwen3-32B | Average Token probabilities |
| 0.5579 | 0.5506 | 0.5653 | TriviaQA | Qwen3-32B | Model confidence |
| 0.6603 | 0.6504 | 0.6691 | TriviaQA | Qwen3-32B | Attention score (LLM-Check) |
| 0.5454 | 0.5355 | 0.5544 | TriviaQA | Qwen3-32B | Average Token Entropy |
| 0.7304 | 0.7221 | 0.7382 | TriviaQA | Qwen3-32B | Hidden State Logistic Regression |
| 0.7304 | 0.7221 | 0.7382 | TriviaQA | Qwen3-32B | Hidden State Logistic Regression |
| 0.8974 | 0.8923 | 0.9024 | TriviaQA | Qwen3-32B | Head Entropy MLP |
| 0.8884 | 0.8830 | 0.8941 | TriviaQA | Qwen3-32B | Head Entropy XGBoost |
| 0.8995 | 0.8946 | 0.9047 | TriviaQA | Qwen3-32B | Head Entropy Logistic Regression |
| 0.7395 | 0.7282 | 0.7509 | HotpotQA | 8B Llama 3.1 | Average Token probabilities |
| 0.5365 | 0.5280 | 0.5451 | HotpotQA | 8B Llama 3.1 | Model confidence |
| 0.5650 | 0.5513 | 0.5783 | HotpotQA | 8B Llama 3.1 | Attention score (LLM-Check) |

| | | | | | |
|---|---|---|---|---|---|
| 0.7533 | 0.7426 | 0.7642 | HotpotQA | 8B Llama 3.1 | Average Token Entropy |
| 0.7756 | 0.7650 | 0.7862 | HotpotQA | 8B Llama 3.1 | Hidden State Logistic Regression |
| 0.7756 | 0.7650 | 0.7862 | HotpotQA | 8B Llama 3.1 | Hidden State Logistic Regression |
| 0.8841 | 0.8768 | 0.8913 | HotpotQA | 8B Llama 3.1 | Head Entropy MLP |
| 0.8540 | 0.8451 | 0.8623 | HotpotQA | 8B Llama 3.1 | Head Entropy XGBoost |
| 0.8731 | 0.8647 | 0.8812 | HotpotQA | 8B Llama 3.1 | Head Entropy Logistic Regression |
| 0.7719 | 0.7591 | 0.7855 | HotpotQA | 3B Llama 3.2 | Average Token probabilities |
| 0.5065 | 0.5035 | 0.5092 | HotpotQA | 3B Llama 3.2 | Model confidence |
| 0.6873 | 0.6704 | 0.7041 | HotpotQA | 3B Llama 3.2 | Attention score (LLM-Check) |
| 0.7733 | 0.7599 | 0.7876 | HotpotQA | 3B Llama 3.2 | Average Token Entropy |
| 0.7776 | 0.7651 | 0.7893 | HotpotQA | 3B Llama 3.2 | Hidden State Logistic Regression |
| 0.7776 | 0.7651 | 0.7893 | HotpotQA | 3B Llama 3.2 | Hidden State Logistic Regression |
| 0.8904 | 0.8821 | 0.8982 | HotpotQA | 3B Llama 3.2 | Head Entropy MLP |
| 0.8619 | 0.8532 | 0.8705 | HotpotQA | 3B Llama 3.2 | Head Entropy XGBoost |
| 0.8816 | 0.8734 | 0.8900 | HotpotQA | 3B Llama 3.2 | Head Entropy Logistic Regression |
| 0.5838 | 0.5704 | 0.5987 | HotpotQA | Qwen3-1.7B | Average Token probabilities |
| 0.5036 | 0.5008 | 0.5063 | HotpotQA | Qwen3-1.7B | Model confidence |
| 0.7021 | 0.6915 | 0.7131 | HotpotQA | Qwen3-1.7B | Attention score (LLM-Check) |
| 0.5986 | 0.5849 | 0.6135 | HotpotQA | Qwen3-1.7B | Average Token Entropy |
| 0.8366 | 0.8277 | 0.8451 | HotpotQA | Qwen3-1.7B | Hidden State Logistic Regression |
| 0.8366 | 0.8277 | 0.8451 | HotpotQA | Qwen3-1.7B | Hidden State Logistic Regression |
| 0.8858 | 0.8786 | 0.8931 | HotpotQA | Qwen3-1.7B | Head Entropy MLP |
| 0.8795 | 0.8720 | 0.8864 | HotpotQA | Qwen3-1.7B | Head Entropy XGBoost |
| 0.8831 | 0.8761 | 0.8903 | HotpotQA | Qwen3-1.7B | Head Entropy Logistic Regression |
| 0.5815 | 0.5690 | 0.5950 | HotpotQA | Qwen3-8B | Average Token probabilities |
| 0.6809 | 0.6725 | 0.6892 | HotpotQA | Qwen3-8B | Model confidence |
| 0.6813 | 0.6680 | 0.6934 | HotpotQA | Qwen3-8B | Attention score (LLM-Check) |
| 0.5580 | 0.5452 | 0.5709 | HotpotQA | Qwen3-8B | Average Token Entropy |
| 0.7691 | 0.7576 | 0.7803 | HotpotQA | Qwen3-8B | Hidden State Logistic Regression |
| 0.7691 | 0.7576 | 0.7803 | HotpotQA | Qwen3-8B | Hidden State Logistic Regression |
| 0.8556 | 0.8467 | 0.8641 | HotpotQA | Qwen3-8B | Head Entropy MLP |
| 0.8414 | 0.8320 | 0.8502 | HotpotQA | Qwen3-8B | Head Entropy XGBoost |
| 0.8529 | 0.8447 | 0.8613 | HotpotQA | Qwen3-8B | Head Entropy Logistic Regression |
| 0.6970 | 0.6820 | 0.7117 | MedMCQA | 8B Llama 3.1 | Average Token probabilities |
| 0.5809 | 0.5650 | 0.5966 | MedMCQA | 8B Llama 3.1 | Model confidence |
| 0.5445 | 0.5262 | 0.5612 | MedMCQA | 8B Llama 3.1 | Attention score (LLM-Check) |
| 0.6965 | 0.6815 | 0.7112 | MedMCQA | 8B Llama 3.1 | Average Token Entropy |
| 0.5806 | 0.5637 | 0.5975 | MedMCQA | 8B Llama 3.1 | Hidden State Logistic Regression |
| 0.7473 | 0.7321 | 0.7613 | MedMCQA | 8B Llama 3.1 | Head Entropy MLP |
| 0.7267 | 0.7114 | 0.7428 | MedMCQA | 8B Llama 3.1 | Head Entropy XGBoost |
| 0.7409 | 0.7261 | 0.7557 | MedMCQA | 8B Llama 3.1 | Head Entropy Logistic Regression |
| 0.6779 | 0.6599 | 0.6940 | MedMCQA | 3B Llama 3.2 | Average Token probabilities |
| 0.5368 | 0.5203 | 0.5538 | MedMCQA | 3B Llama 3.2 | Model confidence |
| 0.5174 | 0.5015 | 0.5339 | MedMCQA | 3B Llama 3.2 | Attention score (LLM-Check) |
| 0.6692 | 0.6514 | 0.6850 | MedMCQA | 3B Llama 3.2 | Average Token Entropy |
| 0.5614 | 0.5449 | 0.5787 | MedMCQA | 3B Llama 3.2 | Hidden State Logistic Regression |
| 0.5614 | 0.5449 | 0.5787 | MedMCQA | 3B Llama 3.2 | Hidden State Logistic Regression |
| 0.7169 | 0.7010 | 0.7322 | MedMCQA | 3B Llama 3.2 | Head Entropy MLP |
| 0.7014 | 0.6865 | 0.7166 | MedMCQA | 3B Llama 3.2 | Head Entropy XGBoost |
| 0.7215 | 0.7056 | 0.7361 | MedMCQA | 3B Llama 3.2 | Head Entropy Logistic Regression |
| 0.7081 | 0.6874 | 0.7279 | MedMCQA | Qwen3-1.7B | Average Token probabilities |
| 0.7922 | 0.7771 | 0.8060 | MedMCQA | Qwen3-1.7B | Model confidence |
| 0.7999 | 0.7855 | 0.8128 | MedMCQA | Qwen3-1.7B | Attention score (LLM-Check) |
| 0.6633 | 0.6412 | 0.6841 | MedMCQA | Qwen3-1.7B | Average Token Entropy |
| 0.8405 | 0.8279 | 0.8520 | MedMCQA | Qwen3-1.7B | Hidden State Logistic Regression |
| 0.8908 | 0.8814 | 0.9002 | MedMCQA | Qwen3-1.7B | Head Entropy MLP |
| 0.8860 | 0.8750 | 0.8953 | MedMCQA | Qwen3-1.7B | Head Entropy XGBoost |
| 0.8976 | 0.8878 | 0.9067 | MedMCQA | Qwen3-1.7B | Head Entropy Logistic Regression |
| 0.5417 | 0.5179 | 0.5628 | MedMCQA | Qwen3-8B | Average Token probabilities |

| 0.8049 | 0.7881 | 0.8227 | MedMCQA | Qwen3-8B | Model confidence |
| 0.8394 | 0.8269 | 0.8520 | MedMCQA | Qwen3-8B | Attention score (LLM-Check) |
| 0.5301 | 0.5073 | 0.5505 | MedMCQA | Qwen3-8B | Average Token Entropy |
| 0.8898 | 0.8798 | 0.8994 | MedMCQA | Qwen3-8B | Hidden State Logistic Regression |
| 0.9408 | 0.9340 | 0.9472 | MedMCQA | Qwen3-8B | Head Entropy MLP |
| 0.9412 | 0.9343 | 0.9478 | MedMCQA | Qwen3-8B | Head Entropy XGBoost |
| 0.9475 | 0.9412 | 0.9539 | MedMCQA | Qwen3-8B | Head Entropy Logistic Regression |
| 0.5630 | 0.5451 | 0.5809 | MedMCQA | Qwen3-32B | Average Token probabilities |
| 0.7393 | 0.7260 | 0.7517 | MedMCQA | Qwen3-32B | Model confidence |
| 0.6884 | 0.6714 | 0.7058 | MedMCQA | Qwen3-32B | Attention score (LLM-Check) |
| 0.6314 | 0.6140 | 0.6482 | MedMCQA | Qwen3-32B | Average Token Entropy |
| 0.7446 | 0.7285 | 0.7604 | MedMCQA | Qwen3-32B | Hidden State Logistic Regression |
| 0.8742 | 0.8633 | 0.8847 | MedMCQA | Qwen3-32B | Head Entropy MLP |
| 0.8778 | 0.8674 | 0.8877 | MedMCQA | Qwen3-32B | Head Entropy XGBoost |
| 0.8751 | 0.8644 | 0.8855 | MedMCQA | Qwen3-32B | Head Entropy Logistic Regression |

## H    SHAPLEY VALUE HEATMAPS

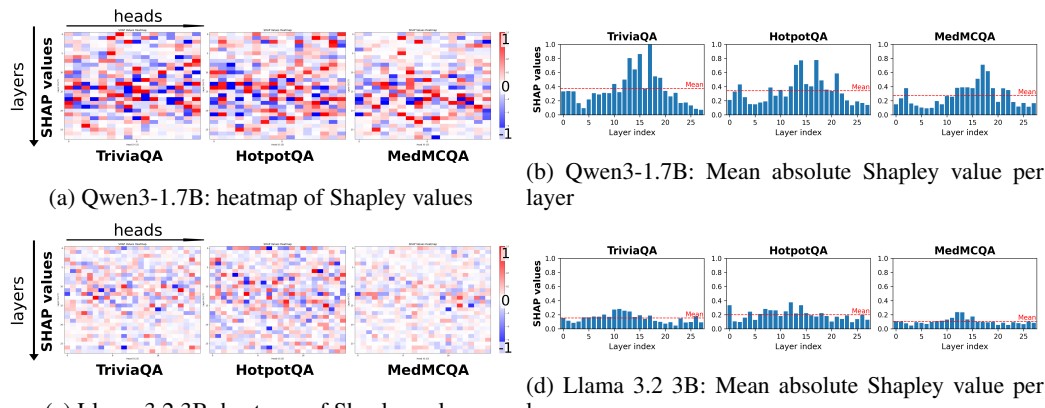

(a) Qwen3-1.7B: heatmap of Shapley values

(b) Qwen3-1.7B: Mean absolute Shapley value per layer

(c) Llama 3.2 3B: heatmap of Shapley values

(d) Llama 3.2 3B: Mean absolute Shapley value per layer

## I    LLM CONTRIBUTIONS

In this work, we used LLMs to help detect writing and grammar errors as well as to create figures.

