# OpenReview forum: "Attention Head Entropy of LLMs Predicts Answer Correctness"
_ICLR.cc/2026/Conference — ICLR 2026 Conference Desk Rejected Submission_

### Official Review · Reviewer_JuPa · 2025-10-19

**Soundness:** 4
**Presentation:** 3
**Contribution:** 2
**Rating:** 2
**Confidence:** 4

**Summary:**

This paper proposes using the attention entropy of each head as features, concatenating them, and training a classifier on top to predict whether a model’s generation is correct or incorrect. The authors evaluate their method on various datasets, including out-of-distribution experiments. They also use Shapley-value attribution to identify which attention heads are most important.

**Strengths:**

- The paper is well written.
- The idea is simple and effective.
- There are good ablation studies and, more importantly, OOD experiments.
- The experiments are comprehensive, and the results are promising

**Weaknesses:**

- The most important weakness of the paper is its novelty. There is already a popular work which uses attention maps to extract features and train a classifier model: https://arxiv.org/pdf/2407.07071. The only difference between this work and the other work is how to extract the feature.
- The data scaling/low data experiments are missing. I would like to see how the performance changes with less or more data.
 - More insights about why this idea works could be helpful. For instance, why is entropy a good feature extractor?

**Questions:**

N/A

---

> ### Author Response · Authors · 2025-11-26
>
> Thank you for the helpful feedback. We will add https://arxiv.org/pdf/2407.07071 (Lookback Lens) to our related work.
>
> **Novelty vs. Lookback Lens:** We acknowledge that prior work has used attention mechanisms and linear models for answer prediction. However, our contribution is demonstrating that entropy as a feature extractor provides a theoretically grounded, interpretable, and effective approach. The following table highlights fundamental differences:
>
> | Aspect | Lookback Lens | Head Entropy | Difference |
> |--------|---------------|--------------|----------------|
> | **Core Concept** | Uses ratio of attention on context tokens vs. newly generated tokens, point esitmation about attention | Uses entropy of attention distributions within each head, measures spread of attention | Fundamentally different signal: ratio vs. information content |
> | **Feature Definition** | LR = context / (context + new) | Entropy | Lookback measures location of attention mass; Head Entropy measures spread of attention |
> | **Problem Focus** | Contextual hallucinations, when model has correct info in context but generates incorrect output | Answer correctness, whether generated answer is correct for QA tasks | Different problem framings |
> | **Interpretability Method** | Not extensively discussed | Shapley values for head attribution | Head Entropy has deeper interpretability analysis |
> | **Key Insight** | Higher lookback ratio (attending more to context) correlates with factuality | Entropy as a summary statistic for attention information content | Different insights |
> | **Datasets** | CNN/DM (summarization), Natural Questions (question without context), sampled examples from test set | TriviaQA (question without context), HotpotQA (answer in context), MedMCQA (MC) on full eval set | Different tasks |
> | **Models Tested** | LLaMA-2-7B-Chat, LLaMA-2-13B-Chat | Qwen3 (1.7B, 8B, 32B), LLaMA 3.1 (8B), LLaMA 3.2 (3B) | Head Entropy tests more models, not only one model class |
> | **Calibration** | Not extensively discussed | ECE < 0.1, extensively analyzed | Head Entropy emphasizes calibration |
> | **Baseline Comparisons** | Hidden states (24th, 28th, 32nd layers), NLI models | Token probability, verbalized confidence, attention scores, hidden states, token entropy | More comprehensive baselines in Head Entropy |
> | **Positive/Negative Heads** | Not discussed | Analyzes heads with positive vs. negative coefficients | Head Entropy provides richer head-level analysis |
> | **Thinking Tokens** | Not applicable (models without CoT) | Analyzes question, think, and answer tokens separately | Head Entropy considers reasoning-oriented models |
> | **Computational Efficiency** | Not analyzed | Less than an extra forward pass, O(m) memory | Both efficient, Head Entropy provides complexity analysis |
>
> **Key Distinction:** Lookback Lens measures where the model attends (a spatial ratio), while HEAD ENTROPY measures how diffuse attention is (an information-theoretic quantity). These capture fundamentally different aspects of attention behavior. A model can have high lookback ratio (attending to context) with either low entropy (focused on specific context tokens) or high entropy (diffused across many context tokens). Both papers demonstrate the utility of attention for prediction, but address different problems with different feature definitions. We will add Lookback Lens as a baseline and expand our comparison in the related work.
>
> **Data Scaling Experiments:**  We appreciate this suggestion. We will include learning curves showing performance vs. training set size to demonstrate data efficiency. Experiments ([Llama 8B data ablation](https://postimg.cc/hJfH1DQD) and [Qwen 8B data ablation](https://postimg.cc/Z06frBnZ)) suggest HEAD ENTROPY maintains strong performance even with 20-30% of training data, but we will provide comprehensive results in revision.
>
> **Why Entropy Works:** Thank you for highlighting this important point. Entropy quantifies attention uncertainty, the fundamental measure of information content in attention mechanisms. Entropy captures how attention spreads across tokens ( Section 4.1). Lower entropy indicates focused, decisive attention patterns that empirically correlate with correct outputs, while higher entropy suggests diffuse, uncertain attention often associated with incorrect reasoning. Appendix C provides empirical evidence for this relationship, analyzing the entropy distribution of the most predictive heads. The analysis shows that incorrect predictions exhibit both higher entropy and higher variance, supporting our theoretical framework. We will strengthen this discussion in our revision with additional analysis of attention patterns in correct vs. incorrect cases.
>
> We hope these clarifications and planned revisions address your concerns. We believe HEAD ENTROPY makes a valuable contribution to uncertainty quantification for LLMs and would be a strong fit for ICLR 2026. We respectfully ask you to reconsider your score.

---

### Official Review · Reviewer_vnso · 2025-10-27

**Soundness:** 2
**Presentation:** 3
**Contribution:** 2
**Rating:** 4
**Confidence:** 4

**Summary:**

The paper introduces attention head entropy as a quantitative metric to measure the uncertainty of LLMs by analyzing their internal attention patterns. Specifically, it computes the Shannon entropy of the normalized attention weights from each attention head, interpreting lower entropy as more focused and confident attention and higher entropy as more diffuse or uncertain behavior. The authors evaluate this metric on 5 instruction-tuned LLMs across 3 question-answering datasets and train logistic regression models to predict output correctness based on the computed entropies. They further use Shapley-value analysis to identify which layers and heads contribute most to prediction performance, providing insights into where reliability-related signals emerge within transformer architectures.

**Strengths:**

1. The paper is clearly written and easy to follow. The proposed method is conceptually straightforward and well-presented.

2. The authors evaluate their approach on five instruction-tuned LLMs and three QA datasets. Results show that head entropy consistently outperforms baseline uncertainty metrics and generalizes well across different model families and sizes, demonstrating both robustness and applicability.

**Weaknesses:**

1. The proposed entropy measure appears highly dependent on the specific query content. Even though entropies are averaged over all tokens within an answer, the resulting entropy-based correctness estimates should still be regarded as query-conditional rather than global indicators of confidence. The model’s attention behavior and therefore its entropy varies strongly with input semantics, which could limit generalization across queries or domains. I appreciate if the authors provide any evidence in resolving my concern here.

2. The authors do not justify why their entropy metric uses only the QK-based attention distribution, nor do they test alternative definitions involving the Value matrix or post-attention activations (for example, the entropy after directly decoding the unembedding matrix using logit lens [1]). Additionally, this way of defining entropy seems not new [2].

3. The proposed metric assumes that lower attention entropy implies higher model confidence, which in turn correlates with correctness. However, large language models are known to sometimes generate highly confident but incorrect outputs. This phenomenon suggests that attention entropy may not be a reliable measure of epistemic uncertainty and can fail in cases of confidently wrong reasoning, especially in hallucination-prone or overfitted regimes.

4. All reported results are associational for correlations between entropy and correctness, without an intervention-based or causal analysis. To me it seems that the results cannot be interpreted as evidence of a causal effect.

5. The division into “question,” “thinking,” and “answer” tokens relies on heuristic, task-specific boundaries that may not generalize across prompts or datasets. It remains unclear whether the approach performs consistently if the prompt format or token segmentation changes (for example, different delimiters or reasoning styles). An ablation over prompt variants would be necessary to verify the robustness of this segmentation-dependent analysis.

[1] Nostalgebraist., “Interpreting GPT: the Logit Lens”

[2] Zhang et al., “Attention Entropy is a Key Factor: An Analysis of Parallel Context Encoding with Full-attention-based Pre-trained Language Models” https://arxiv.org/abs/2412.16545

**Questions:**

The paper does not discuss the distribution of correct versus incorrect answers in the evaluation datasets. If correctness labels are imbalanced, the logistic regression classifier could be biased toward the majority class, leading to inflated or misleading performance metrics.
It would be important to report the base accuracy of the underlying model on these datasets and whether imbalance exists.

---

> ### Author Response · Authors · 2025-11-26
>
> Thank you for your thoughtful feedback.
>
> **Dependence on specific query content:** HEAD ENTROPY predicts the likelihood of correctness for each query-answer pair (e.g., >0.9 AUC for TriviaQA). We aim to assess per-instance reliability rather than global model confidence. Global estimates can be derived by averaging across queries.
>
> **Entropy metric:**
>
> *Why QK-based attention entropy:* The attention distribution represents uncertainty over which tokens to attend to, independent of what information is retrieved. Entropy directly quantifies this allocation uncertainty [11]. Including the Value matrix or post-attention activations may conflate attention allocation with semantic information flow.
>
> *Regarding alternatives:* Entropy "after decoding via logit lens" measures output distribution uncertainty, not attention distribution uncertainty, which captures different aspects of model computation. Our work investigates attention mechanism behavior, for which QK-based entropy is the appropriate measure. Post-attention measurements would introduce confounds from subsequent layers, MLPs, and residual connections.
>
> *On novelty:* We will cite "Attention Entropy is a Key Factor". Our contribution is systematically analyzing entropy dynamics across generation and demonstrating its utility for predicting answer correctness. The novelty lies in these empirical findings and their implications for uncertainty estimation.
>
> *Additional measures:* We will expand related work to discuss Value-based and post-attention alternatives.
>
> **Entropy and overconfidence:** We do not assume lower entropy implies higher confidence. Instead, we empirically test whether HEAD ENTROPY predicts answer correctness. Our calibration analysis (Figure 3) shows HEAD ENTROPY achieves strong calibration (ECE <0.1), with likelihood estimates that match actual correctness rates.
>
> **Empirical results:** This paper combines theoretical motivation (Section 4.1) with empirical validation. HEAD ENTROPY predicts answer correctness with up to 0.95 AUC across models and datasets.
>
> **Division into "question," "thinking," and "answer":** We focus on instruction-tuned models as they represent the primary deployment setting for QA applications, currently the predominant use case for contemporary LLMs. Importantly, HEAD ENTROPY performs well across datasets with different context structures:
>
> - **TRIVIAQA:** Question includes the "Question"; Answer includes the "correct open-ended answer"
> - **HOTPOTQA:** Question includes the "Question, context"; Answer includes the "correct open-ended answer"
> - **MEDMCQA:** Question includes the "Question, context and answer options"; Answer includes the "correct answer from the choices"
>
> We use standard dataset-specific prompt designs (Appendix A.1-3).
>
> **Task-specific vs. task-invariant heads:** Our experiments investigate whether predictive head weightings generalize across tasks. Table 2 shows task-specific patterns: training on one dataset yields 5-35% AUROC drops on others, indicating domain-dependent head importance. This aligns with [3] and suggests attention heads serve different roles across tasks. Joint training recovers near-in-domain performance (drops <2% AUROC), demonstrating that a single model can learn multiple task-specific weightings. In addition, please refer to the "OOD evaluation:" section of reviewer EetY for additional experiments. We will expand this discussion in revision.
>
> **Correct versus incorrect answers:** Thank you for raising this important point. Model accuracies on our evaluation datasets range from 0.33 to 0.69 (see Appendix F), resulting in reasonably balanced correct/incorrect splits. Additionally, we primarily report AUROC, which is robust to class imbalance compared to accuracy-based metrics. We will add a brief note on class balance in the main text for clarity.
>
> [11 ] Claude E Shannon. "A mathematical theory of communication." The Bell system technical journal, 27(3):379–423, 1948.

---

### Official Review · Reviewer_3nKc · 2025-10-28

**Soundness:** 3
**Presentation:** 3
**Contribution:** 2
**Rating:** 4
**Confidence:** 3

**Summary:**

This paper proposes HEAD ENTROPY, a white-box method for predicting LLM correctness during inference. The key insight is that certain attention heads exhibit distinct entropy patterns when models generate correct vs. incorrect answers. Using per-head entropies as features for sparse logistic regression, the method achieves 0.07-0.15 AUROC improvements over baselines on 5 instruction-tuned LLMs and 3 QA datasets. Shapley value analysis reveals that middle-layer attention heads are most informative, providing mechanistic insights into model failures.

**Strengths:**

1. Strong Empirical Results: The method consistently outperforms baselines by meaningful margins across diverse QA tasks (TriviaQA, HotpotQA, MedMCQA) and multiple model families (Qwen, Llama). The 0.07-0.15 AUROC improvements are substantial.

2. Practical Efficiency: The approach adds minimal computational overhead (negligible compared to LLM inference) while requiring only a single forward pass. This makes it genuinely deployable in real systems.

3. Mechanistic Interpretability: Using Shapley values to identify that middle-layer heads contribute most to correctness prediction is insightful and could inform model design and debugging. The finding is consistent across architectures.

**Weaknesses:**

The evaluation scope is a significant limitation. The paper only evaluates on three QA datasets focusing on factual retrieval, which limits the generalizability of findings. More concerning, the experiments are restricted to instruction-tuned models, leaving unclear whether the approach works equally well for base models or other architectures like mixture-of-experts or retrieval-augmented models. The medical domain (MedMCQA) performance is notably weaker, showing only 0.05 AUROC improvement compared to 0.07-0.15 on other datasets, yet the paper provides limited discussion of why certain domains are more challenging for this approach.

The theoretical foundations underlying the method lack depth. The paper doesn't adequately explain why middle-layer heads are specifically informative for correctness prediction—it identifies that they are empirically, but the mechanistic reason remains unclear. More fundamentally, it's uncertain how the relationship between attention entropy and answer correctness emerges during training. The contributions of pre-training versus instruction-tuning to this phenomenon are not investigated, and there's minimal analysis of what actual patterns these supposedly important heads learn or attend to.

Cross-dataset transfer results reveal a worrying pattern: performance drops between 5-35% when models trained on one dataset are evaluated on another, suggesting that task-specific patterns dominate over general correctness indicators. While the paper shows that training on all datasets together improves performance, it doesn't clarify which aspects of the approach generalize and which remain task-specific. Additionally, there's no evaluation on adversarial inputs, out-of-distribution examples, or scenarios with significant domain shift, which would test the robustness of the method in challenging real-world conditions.

Several methodological choices lack justification. The use of sparse logistic regression for classification isn't motivated—no comparison with other classifiers is provided to demonstrate why this is optimal. The paper shows limited ablation on the L1 regularization parameter, and the decision to aggregate entropy over tokens by averaging rather than exploring other aggregation schemes seems arbitrary without empirical justification.

**Questions:**

1. Why do middle-layer heads specifically capture correctness signals?

2. How does HEAD ENTROPY perform on out-of-distribution inputs or adversarial examples? Does the relationship between entropy and correctness hold?

3. Why is performance substantially worse on MedMCQA?

4. Have you tested the method on base models (non-instruction-tuned)?

5. Can section-level aggregation be justified theoretically or empirically compared to alternatives?

6. What happens with models that use different attention mechanisms (e.g., GQA, rotary embeddings)?

---

> ### Author Response · Authors · 2025-11-26
>
> Thank you for your thorough review and thoughtful feedback.
>
> **Evaluation scope limitation:** We appreciate your attention to evaluation comprehensiveness. Our study includes 15 model-dataset pairs evaluated against 6 baselines on both exact match and F1 metrics, plus calibration performance, totaling 270 comparisons. This spans 4 model sizes (1.7B-32B parameters), 2 families (Llama and Qwen), 3 QA tasks, and 2 inference modes (greedy and sampling). Notably, Qwen3 models are mixture-of-experts architectures, demonstrating HEAD ENTROPY's applicability beyond dense models. We focus on instruction-tuned models as they represent the primary deployment setting for contemporary QA applications. We agree that extending to base models and RAG systems represents valuable future work and will note these as promising directions in our conclusion.
>
> **MedMCQA performance:** We analyse performance on the training set of MedMCQA by sampling the held out set from the training set.
>
> **Llama-3B**
>
> |Method|Test|Held-Out|
> |---|---|---|
> |Avg Token Prob|0.68|0.74|
> |Model Conf|0.54|0.56|
> |Attn Score|0.52|0.51|
> |Avg Token Ent|0.67|0.74|
> |Hidden State LR|0.56|0.61|
> |Head Ent MLP|0.72|0.78|
> |Head Ent XGB|0.70|0.78|
> |Head Ent LR|0.72|0.79|
>
> **Llama-8B**
>
> |Method|Test|Held-Out|
> |---|---|---|
> |Avg Token Prob|0.70|0.78|
> |Model Conf|0.58|0.57|
> |Attn Score|0.54|0.54|
> |Avg Token Ent|0.70|0.78|
> |Hidden State LR|0.58|0.64|
> |Head Ent MLP|0.75|0.84|
> |Head Ent XGB|0.73|0.82|
> |Head Ent LR|0.74|0.84|
>
> This experiment shows that the performance significantly improves when evaluated on a held out set from the training distribution, suggesting that the test set of this data set is sufficiently OOD. We discuss the OOD results further with reviewer EetY.
>
> **Cross-dataset transfer results:** Thank you for your interest in Table 2. We designed these experiments to test whether head weightings are task-invariant (e.g., heads 32, 7 always important) or task-specific (e.g., heads 58, 2 for medical QA vs. 69, 10 for Trivia). Results show task-specific patterns: single-dataset training yields 5-35% drops cross-domain, aligning with [3]. Crucially, joint training recovers near-in-domain performance (drops <2%), showing one regressor can learn multiple task-specific weightings simultaneously. We will expand this discussion in our revision.
>
> **Theoretical foundations:**
> We appreciate your interest in the theoretical grounding. The differential entropy $\mathrm{H}[\boldsymbol{f}] \doteq \mathbb{E}\_{p(\boldsymbol{f})}[-\log p(\boldsymbol{f})]$ quantifies uncertainty, where $-\log p(\boldsymbol{f})$ measures information content. Applied to attention, entropy captures focus vs. dispersion: high entropy indicates uncertainty about which tokens receive attention. As information theory's foundational uncertainty measure, entropy provides a principled indicator of attention reliability [10]. Middle layers' importance for interpretability has been documented [8,9]. While our empirical results demonstrate entropy's predictive value for answer correctness, we agree that establishing deeper theoretical connections between attention entropy dynamics and model reliability represents an important direction for future research.
>
> **Empirical relationship between attention entropy and answer correctness during training:**
> Thank you for this thoughtful question. We focus on instruction-tuned models as they represent the primary deployment setting for QA applications. Appendix C analyzes the entropy distribution of the most predictive heads, showing that incorrect predictions exhibit both higher entropy and higher variance, providing empirical evidence for the entropy-correctness relationship.
> Regarding base model training dynamics: analyzing how attention entropy evolves during pretraining is indeed an interesting direction. However, this would require a fundamentally different experimental framework with reference-free correctness metrics, as supervised ground truth correctness of answers isn't available during pretraining. We believe these conceptual differences warrant dedicated future work and would be happy to discuss this further if helpful.
>
> **Ablations:** Thank you for raising this point. We do compare linear regression against MLP and XGBoost in Figure 2, where linear regression performs equally well despite being simpler. Our focus is on demonstrating the methodological contribution rather than extensive hyperparameter optimization, so we used standard settings (lambda=1 as default). Please see the ablations here for additional details in the figure links: [Llama-8B c ablation](https://postimg.cc/2VSXq3zf) and [Qwe3-8B c ablation](https://postimg.cc/Z06frBnZ)
>
> We hope we addressed all your concerns to the fullest and hope this is reflected by an increase in your score.
>
> [8] Sriramanan, Gaurang, et al. "Llm-check"
>
> [9]  https://transformer-circuits.pub/2025/attribution-graphs/biology.html
>
> [10] https://arxiv.org/abs/2412.16545

---

### Official Review · Reviewer_EetY · 2025-10-30

**Soundness:** 3
**Presentation:** 2
**Contribution:** 2
**Rating:** 4
**Confidence:** 4

**Summary:**

This paper proposes a lightweight method to build LLM answer checkers for factual questions. The authors investigate the entropy of the multi-head attention logits for modern Transformer-based LLMs and demonstrate that with a simple linear logistic regressor, the attention entropy can be a strong cue to predict whether or not the LLMs' answers are correct or not. Extensive experiments demonstrate the effectiveness and interpretability of the proposed method.

**Strengths:**

- The idea is quite simple, applicable to all Transformer-based LLMs.
- The writing is good with clear definitions and terminology.

**Weaknesses:**

- About entropy calculation
  - In Sec. 4.1, I cannot see how to process the attention entropy of different layers and different sections, and thus, I'm curious about the specific shape of $H$ $ in Equ. 8.
- About generalizability and applicability:
  - This paper focuses only on closed-ended factual questions with ground truth answers, and thus, we can train a binary logistic regressor. It suggests that this method cannot generalize to open-ended QA like chatting, and thus, is not as general as reward modeling.
  - I'm wondering whether this method is only applicable for answer-extraction QA (like SQuAD), where answers should first appear in the context during responding.
  - To demonstrate that, an applicable experiment is to 1) report the percentage of samples where the answers first appear in the responses, either thinking or answers (this will be high), and then 2) the correlation between the positions of the appearance of GT answers in the context and the positions with high entropy values.

- About experiments:
  - In Sec 4.3, although light-weighted, we still need to first finish the whole inference procedure before we can get the correctness prediction, and thus, in Sec. 6, we should also compare with reward models (e.g., LLM-as-a-judge).
  - In Sec. 5, on all three evaluated datasets, we conduct an in-distribution setting (e.g., train on TrivalQA and evaluate on TrivalQA). I would like to see more OoD evaluation, like training on general datasets like TriviaQA and HotpotQA, and then evaluating on domain-specific datasets like MedMCQA.
  - In Sec. 6.6, the usage of Shapley values is good, but the conclusion that mid-layers matter more cannot be a valid conclusion for the interpretability of the proposed method.

**Questions:**

- In lines 76 and 80, the first characters should be capitalized.
- Do not use different definitions of subscripts for the same RV. In Equation 5, the subscript of $H$ means head index, while in Equation 6, it means sample index.

---

> ### Author Response · Authors · 2025-11-26
>
> Thank you for your thoughtful review and constructive feedback. We really appreciate your time you spent on this review.
>
> **About entropy calculation:** The shape of $\overline{\mathrm{H}}\_{\mathrm{i}}$ in Equation 8 is $\overline{\mathrm{H}}\_{\mathrm{i}} \in \mathbb{R}^{\mathrm{m}}$, where each $\overline{\mathrm{H}}\_{k, i} \in \mathbb{R}$ represents one scalar per head.
>
> **About generalizability and applicability:** HEAD ENTROPY applies wherever correctness is measurable, via exact match, F1, GPT-4o labels, BERTScore, or human preferences. For contexts without measurable correctness (e.g., creative tasks), HEAD ENTROPY is less applicable, but so is the notion of "correctness" itself. HEAD ENTROPY is not limited to extractive QA: TriviaQA (answers not in questions) and HotpotQA (answers in context) both show strong performance (Figure 2). These results provide real-world evidence that HEAD ENTROPY generalizes beyond answer-extraction QA. We will clarify the purposes of each of the selected datasets in the experimental setup section.
>
> **About experiments:** We appreciate the suggestion to compare with LLM-as-a-judge approaches. However, these methods address fundamentally different use cases: HEAD ENTROPY is a white-box method leveraging internal model representations, while LLM-as-a-judge is a black-box approach requiring separate evaluation calls. We focus our comparisons on white-box uncertainty methods that share HEAD ENTROPY's design philosophy:
>
> *Compute and data requirements:* LLM-as-a-judge and HEAD ENTROPY have fundamentally different deployment requirements. Judge models require either commercial API access or substantial inference compute for calibration and debiasing [1], while fine-tuned judges need full model weights yet offer limited generalization [2]. HEAD ENTROPY, by contrast, uses only training data and model predictions to provide calibrated correctness estimates.
>
> *Complementary:* LLM-as-a-judge offers external evaluation capabilities, while HEAD ENTROPY provides interpretable, white-box uncertainty quantification directly from internal representations, particularly valuable in high-stakes domains requiring transparency.
>
> **OOD evaluation:** Thank you for your interest in Table 2. We designed these experiments to investigate whether the predictive head weighting learned by our logistic regression is task-invariant or task-specific. Task-invariant would indicate universal heads consistently predictive of correctness across domains (e.g., heads 32 and 7 always being important). Task-specific would show domain-dependent head importance (e.g., heads 58, 2 for medical QA vs. heads 69, 10 for Trivia QA).
>
> Table 2 demonstrates that head importance is predominantly task-specific: training on one dataset yields 5-35% AUROC drop on others, but joint training across all datasets recovers near-in-domain performance (drop <2% AUROC). The correlation table below confirms this: coefficients trained on single datasets weakly correlate, while jointly-trained coefficients strongly correlate, showing a single regressor can learn multiple task-specific weightings simultaneously.
> Llama 8B:
> || All datasets | TriviaQA | HotpotQA | MedMCQA |
> |--------------|--------------|----------|----------|---------|
> | All datasets | 1 ||||
> | TriviaQA | 0.408 | 1 |||
> | HotpotQA | 0.605 | 0.124| 1 ||
> | MedMCQA | 0.374| 0.035 | 0.057| 1 |
>
> A visualisation of the log coefficient can be found here: [coefficients-llama8b.png](https://postimg.cc/CBZcbTjj)
> The visualization shows that coefficients cluster without correlation when trained on single datasets, but exhibit clear trends when trained jointly (colormap). We will add this analysis for all models and expand the discussion of task-specific vs. task-invariant patterns in our revision.
>
> **Interpretability:** To enhance interpretability, we provide: 1) visualizations showing the predictiveness of each layer with Shapley values. Many uncertainty estimators rely on final token probability distributions [4-5], but post-trained LLMs are often miscalibrated [7]. HEAD ENTROPY extracts signals from middle layers, which better indicate answer correctness. (2) Appendix C, which analyzes the entropy distribution of correct vs. incorrect of the most predictive heads via Shapley values. This analysis reveals that incorrect predictions exhibit both higher entropy and higher variance in attention head entropy. (3) visualizations of task-specific and task-invariant head weights of the linear regression model (see above).
>
> We hope these revisions address your concerns and would appreciate your reconsideration of the score.
>
> [1] https://arxiv.org/html/2410.13341v1
> [2] https://aclanthology.org/2025.findings-acl.306/
> [3] https://arxiv.org/abs/2409.03752
> [4] https://arxiv.org/abs/2307.10236
> [5] https://aclanthology.org/W18-6322/
> [6] https://aclanthology.org/2023.findings-acl.93/
> [7] https://arxiv.org/abs/2409.19817

---

> > ### Comment · Reviewer_EetY · 2025-11-27
> >
> > Thank the authors for the responses.
> >
> > I believe that we have all witnessed the empirical effectiveness of the proposed method, particularly in **in-distribution** settings. However, my main concerns stand, which have also been raised by **ALL** four reviewers, that the underlying reasons **why this method works** are still unclear (and thus, we care about implementations, generalizability, and other details).
> >
> > A theoretical analysis will be the best, while a statistical analysis of the head entropy sequence distribution will also be beneficial. For example, you might check when the prediction is correct, does the input sequence of head entropy contain several value peaks on a small fraction of positions (i.e., saliency detection) or generally high or low (i.e., pattern recognition)? And what are the results for wrong predictions?
> >
> > The conclusion above might help explain your technical choices, like why middle layers perform best. And I still do not understand why you take the average with respect to the output positions but not the head dimension, since both factors might produce totally different attention maps in practice.
> >
> > And I wish this statistical analysis could perform as a basis to help you answer the questions raised by other reviewers :)

---

> > > ### Author Response · Authors · 2025-12-01
> > >
> > > Thank you! We too are interested in this question, but felt the fact that the empirical results compel us to share the method now, especially when combined with the fact the analysis is already more thorough than prior works in the space.
> > >
> > > In order to aid future progress, we propose releasing a dataset of the attention head entropies. We will also create an additional appendix with statistical investigations including tracking the evolution of the head entropy over training (evaluated on the OMLO model intermediate checkpoints) and a clustering analysis. So far the characterization of the decision boundary (Figures 2, 3) and Shapley value analyses that are already in the paper are the most informative in our opinions.
> > >
> > > We wholeheartedly agree that this is an important future direction and look forward to both investigating ourselves and seeing the ideas the community brings.

---

### Note · Program_Chairs · 2026-01-17
**Submission Desk Rejected by Program Chairs**

The following references in this submission do not refer to real documents and/or have major errors in bibliographic information:

 Shehzaad Dhuliawala, Christopher J Maddison, Joshua Maynez, William Saunders, Jonathan Uesato, Jan Leike, Joel Z Leibo, Sebastian Borgeaud, Laurent Sifre, Amelia Norelli, et al. Chain-of-verification reduces hallucination in large language models. arXiv preprint arXiv:2310.04373, 2023.